# Uncovering Patterns in Dairy Cow Behaviour: A Deep Learning Approach with Tri-Axial Accelerometer Data

**DOI:** 10.3390/ani13111886

**Published:** 2023-06-05

**Authors:** Paolo Balasso, Cristian Taccioli, Lorenzo Serva, Luisa Magrin, Igino Andrighetto, Giorgio Marchesini

**Affiliations:** Dipartimento di Medicina Animale, Produzioni e Salute, Università degli Studi di Padova, 35020 Legnaro, Italylorenzo.serva@unipd.it (L.S.);

**Keywords:** convolutional neural network, machine learning, precision livestock farming, animal welfare

## Abstract

**Simple Summary:**

The early detection of behavioural changes based on variations in dairy cows’ daily routines is of paramount importance to the timely identification of the onset of disease. However, the effectiveness in identifying these changes through the use of sensors is dependent on the accuracy and precision of the system used. This study tested the performance of deep learning models in classifying the behaviour of dairy cows on the basis of the data acquired through a tri-axial accelerometer. The results were compared with those obtained from the same raw data analysed by classical machine learning algorithms. Among the tested models, an 8-layer convolutional neural network showed the highest performance in predicting the considered behaviours.

**Abstract:**

The accurate detection of behavioural changes represents a promising method of detecting the early onset of disease in dairy cows. This study assessed the performance of deep learning (DL) in classifying dairy cows’ behaviour from accelerometry data acquired by single sensors on the cows’ left flanks and compared the results with those obtained through classical machine learning (ML) from the same raw data. Twelve cows with a tri-axial accelerometer were observed for 136 ± 29 min each to detect five main behaviours: standing still, moving, feeding, ruminating and resting. For each 8 s time interval, 15 metrics were calculated, obtaining a dataset of 211,720 observation units and 15 columns. The entire dataset was randomly split into training (80%) and testing (20%) datasets. The DL accuracy, precision and sensitivity/recall were calculated and compared with the performance of classical ML models. The best predictive model was an 8-layer convolutional neural network (CNN) with an overall accuracy and F1 score equal to 0.96. The precision, sensitivity/recall and F1 score of single behaviours had the following ranges: 0.93–0.99. The CNN outperformed all the classical ML algorithms. The CNN used to monitor the cows’ conditions showed an overall high performance in successfully predicting multiple behaviours using a single accelerometer.

## 1. Introduction

In the dairy farming sector, many efforts are being made to meet the production and economic goals of farmers, reduce environmental impacts and preserve animal welfare and health [1]. The health of dairy cows is positively associated with milk yield and profitability. However, although prevention of infectious diseases such as bovine rhinotracheitis and bovine viral diarrhoea through vaccination is readily available, metabolic diseases, mastitis and lameness remain a challenge for farmers [1,2,3]. Diseases affect animal health and welfare, reduce milk quality and production and increase farm management costs. For these reasons, early detection and treatment of sick cows is of great importance since it improves disease prognoses and reduces drug intake [4]. Among the most common early signs of a disease, a change in time spent performing different behaviours is one of the most informative and commonly used. Behaviour changes associated with sickness can appear before the actual clinical symptoms of a disease. It is generally characterized by lethargy, which alters the animal’s normal time budget, increasing resting and lying time and reducing time spent on other activities, including feeding, drinking and interaction with conspecifics and farm personnel [4,5]. Some authors have found that mastitis and lameness in dairy cows are associated with changes in rumination time and physical activity, feeding patterns and periods spent lying down [6,7,8,9]. In summary, analysing data on the physiological behaviours of healthy cows in a herd provides an opportunity for farmers and veterinarians to quickly identify deviations from normal behaviour. Measuring different behaviours, such as lying, resting, feeding, ruminating and others, also provides significant information on animal welfare and the effectiveness of animal management procedures [10,11]. Today, there are commercially available sensors, such as tri-axial accelerometers [12] and artificial intelligence tools, that provide the opportunity to routinely collect and analyse such data, returning predictions of the duration of the different behaviours performed [13]. Suitable effectiveness in identifying an event, such as the onset of a disease through the use of technology, is, however, highly associated with the accuracy and precision of the system used. In cases of low accuracy, the deviations from the behavioural routine identified by the system could be due to a mistaken behaviour prediction.

The reliability and overall accuracy of such predictions depend on many factors, such as the type and sensitivity of the sensors used, the number and location of the sensors on the cow [14,15], the kind and the number of behaviours investigated and, obviously, the predictive models used [9]. To date, the models used in the prediction of a wide range of events (e.g., oestrous, lameness, mastitis, ruminal acidosis, etc.) and behaviours in cows, starting from tri-axial accelerometer data [6,9,12,14,15] or data of other origins, such as the animal’s location [4], were based on classical machine learning (ML), with only a few papers reporting the use of deep learning models [16,17,18,19]. ML methods attempt to find meaningful relationships between features or variables through the detection of hidden patterns among them [2]. Distinct prediction models are based on distinct learning and/or training strategies that affect their overall accuracy [2]. Classical ML prediction systems are based on various stages, such as preprocessing (e.g., de-noising, filtering), feature engineering (time and frequency domain descriptors), feature selection and classification algorithms [20]. Among the latter, especially with regard to behaviour prediction, support vector machine (SVM), the extreme boosting algorithm (XGB), random forest (RF), k-nearest neighbours (KNN) and artificial neural networks are commonly used [4,9,20].

Although classical ML systems have been demonstrated to provide an overall acceptable performance in predicting behaviours [9,14,15], in the feature engineering stage, there is a manual extraction of features that can potentially lead to the loss of some meaningful information [20].

In contrast with classical ML, DL models, also known as deep neural networks, do not need handcrafted feature extractor designs but automatically capture complex features; they consider nonlinearity in the feature space, and they are able to extract complex patterns and spatial or temporal dependencies from the underlying raw data streams [20,21]. The performance of DL models heavily depends on the internal architecture of their algorithms, which can be described, for example, as recurrent neural networks (RNNs), convolutional neural networks (CNNs) and long short-term memory networks (LSTMs) [20,21,22]. DL models were initially created in the 1990s for computer vision applications but since then they have been applied to miscellaneous domains such as self-driving cars, finance and even livestock farming [20,21,23]. In particular, LSTM, CNN and the hybrid form, CNN-LSTM, have become very common for classifying and predicting time series. The LSTM model is a special form of RNN that provides feedback to each neuron and overcomes the vanishing gradient problem of the RNN model through internal loops that maintain only the useful information. The CNN-LSTM combines CNN and LSTM characteristics. First, the CNN extracts important information from the input data and reorganizes the univariate input data into multidimensional batches using convolution. In the second phase, the reorganized batches are used as input into the LSTM. Although DL models in animal production have been recently applied to computer vision with the aim of identifying, for example, individuals [21] or behaviours [24], to our knowledge, their application to tri-axial accelerometry data for behaviour classification is scarce and mainly applied to humans [20,25], with a few exceptions in which they were applied to cattle [18,19,22]. Furthermore, none of the latter studies compared the outcomes of the DL models with those of the classical ML models. The aim of this study is to compare the performance of a CNN in classifying the behaviour of healthy dairy cows based on the data acquired through a tri-axial accelerometer with the results obtained from the same raw data through the use of classical ML models [26].

## 2. Materials and Methods

### 2.1. Ethical Statement

The trial was carried out in accordance with D.Lgs. 26/2014 and EU Directive 2010/63/EU concerning experiments on animals and was endorsed by the animal welfare committee (Organismo Preposto al Benessere Animale committee—OPBA—official number 167326) of Padova University. The experimental protocol was approved by the licensing committee OPBA (official number 167326) of Padova University according to D.Lgs. 26/2014. Furthermore, all methods were performed in accordance with the OPBA’s guidelines and regulations in compliance with D.Lgs. 26/2014. The research adhered to all aspects of the ARRIVE guidelines to conduct both the study design and reporting. The protocol consisted of 27.3 h of observation of 12 healthy, randomly chosen mid-lactating dairy cows wearing single tri-axial accelerometers to assess the accuracy of a deep learning model in predicting cows’ behaviour.

### 2.2. Data Collection

Animal husbandry and data collection are briefly represented in Figure 1 and described in detail by Balasso and colleagues in a paper reporting the use of classical ML to classify dairy cow behaviour [26]. Briefly, the trial was carried out on an Italian dairy farm raising Italian Red-and-White cows in loose housing conditions. Italian Red-and-White is the third most raised dairy cow breed in Italy after Italian Holstein and Italian Brown. It is characterized by a fair average milk production (7146 kg per lactation), good longevity (2.97 lactations) and high carcass quality. Twelve randomly selected healthy cows with 2.87 ± 0.91 lactations and 180 ± 35 days in milk (average ± SD) were observed by trained personnel for an average of 136 ± 29 min per cow over a period of 12 days. The animals were observed approximately between 1100 h and 1500 h, to include the highest variety of behaviours possible, by two trained operators who recorded cow behaviour in real time using Microsoft Excel 2010 (Microsoft, Remond, WA, USA) on a computer synchronized with the sensor. Inter-observer reliability, based on Cohen’s kappa [27], was 0.91. During the observation sessions, each cow was wearing a tri-axial (X, Y, Z) accelerometer (model MSR145 W, PCE Italia s.r.l, Capannori, LU, Italy), applied to the centre of the left side paralumbar fossa with an elastic band, which was kept in position by glue [26]. The sensor was set to collect data at a frequency of 5 Hz [26] to identify short-term behaviours and at the same time save battery life. The accelerometer was fixed to a standing animal’s X, Y and Z axes in a preset position, which was X vertical, Y parallel to the ground and Z orthogonal to the cow’s flank. Five behaviours were considered: moving, standing still, feeding, ruminating and resting, as reported in Table 1.

### 2.3. Dataset Preparation

Acceleration data on the X, Y and Z axes were exported as CSV files using the software program MSR 5.12.04 (PCE Italia s.r.l., Capannori, LU, Italy). Data were then imported into Excel 2010 (Microsoft, Redmond, WA, USA), where the collection time (date, h, min, s, hundredths of a second); acceleration values on the X, Y and Z axes; and the corresponding behaviours were reported for each row in different columns. Statistical analyses were performed using R, Version 3.2.1 (R Core Team 2013). Tri-axial accelerations were recorded every 0.2 s, corresponding to 27.3 h of observation (*n* = 490,900). The observations during which a behaviour was unclear were excluded from the dataset, leaving 25.4 h of observation suitable for analysis (*n* = 456,730), including feeding (4.68 h; *n* = 84,206), moving (4.69 h; *n* = 84,400), resting (7.84 h; *n* = 141,055), ruminating (2.98 h; *n* = 53,744) and standing still (5.18 h; *n* = 93,325).

A list of metrics was obtained based on a rolling window of 15 observations. These metrics are the standard deviation (sd), average (avg), percentage change between an observation and the previous one and the binary value related to it (if the percentage change is negative, the value given is 0; otherwise, it is 1) applied to the X, Y and Z acceleration data for a total of 15 variables. As reported in Figure 2, each interval of 40 observations (8 s), with a sliding interval of 13 observations (33%), was aggregated into one observation unit and associated with a specific behaviour, obtaining a dataset of 211,720 observation units and 15 columns. The 8 s interval was chosen because it offered the best compromise in differentiating one very short behaviour, such as walking, from others.

As reported in Figure 3, to build up a predictive model, the dataset was randomly split into training (80% of the observations, *n* = 169,376) and testing (20% of the observations *n* = 42,344) datasets. The latter was used to estimate the performance of the model. All variables were normalised by considering the mean and the standard deviation of the training dataset. The training dataset is 3-dimensional (169, 362, 40, 15). In fact, each behaviour (y training vector) was associated with an array with a shape (1, 40, 15), where 40 is the timeframe range considered, and 15 was the number of aforementioned variables. The batch size was set to 32; the batch was a subset of the training data used in each iteration of the training algorithm in ‘mini-batch gradient descent’. Thus, in each epoch, the network weights would be updated 5293 times (169,362 over 32 times).

### 2.4. Data Modelling

The data modelling started with the use of a CNN characterized by a 6-layer CNN consisting of 2 convolution, 1 dropout, 1 max pooling, 1 flattening and 1 dense layer and achieved an overall accuracy of 0.76. The CNN was then modified by adding a second dense layer in a second model to eventually reach maximum performance with the third and final 8-layer CNN, which is described below. To further improve performance, a CNN-LSTM was tried; however, it failed to outperform the last CNN, achieving an overall accuracy of 0.91, as reported in Table 1 (values in brackets).

Table 2 shows the structure of the best-performing 8-layer CNN model, which was built using the following layers: 3 convolution layers, 1 dropout layer, 1 max pooling layer, 1 flattened layer and 2 dense layers.

–Convolution is a process in which a small matrix (the kernel or filter) is slid across the input dataset and is transformed on the basis of the filter values. As reported in Table 3, in the Conv1d_1, Conv1d_2 and Conv1d_3 layers, the filters were set at 128, 64 and 32, respectively. For all three layers, the kernel size was set at 3, and the activation function used was the rectified linear unit (RELU). We set padding = ‘valid’ so that the size of the feature maps would gradually decrease as it went through the convolutional layers, which is the default setting option in Keras. Otherwise, ‘Zero Padding’ means filling two edges of each layer’s inputs with zero to keep the size of the inputs and outputs the same. The stride parameter is the number of pixels that a filter moves by once it is inside an input. If it is one, the filter moves right, one pixel at a time. We made the stride parameters equal to one for the convolutional layers and to the same value as the pool size for the pooling layers. If the values of the stride and pooling kernel size are the same, each kernel is prevented from being overlapped.–The dropout layer randomly selects neurons that are ignored during training. This helps to prevent overfitting. To accomplish this, a rate frequency is adopted at each step. In this model, the rate was set to 0.3.–Max pooling was used to reduce the size of the tensor and to accelerate calculations. It downsamples the input representation by calculating the largest value over the window as defined by pool size, which in our case was set to 2. We maintained stride and padding parameters equal to those of the convolution layers.–The flattened layer reduces the data into an array so that the CNN can read it by removing every dimension but one. As reported in Table 3, the output shape of the layer is 544, which is equal to 17 times 32, the two dimensions of the previous layer.–The dense layer consists of a finite number of neurons (mathematical functions) that receive one vector as input and return another vector as output. The first dense layer was made of 100 neurons with a ‘RELU’ activation function and was connected to the last dense layer with a softmax activation function and a length of 5, which is equal to the number of activities to be classified by the model. The model was deployed in Python using Keras [28] with a TensorFlow backend.–It is noteworthy that the final layer’s output shape is 5, given that there are 5 behaviours to classify.

**Table 2 animals-13-01886-t002:** Summary of the deep learning model’s architecture, with a description of the layers used, output shape and the number of parameters and hyperparameters used in the model for each layer.

Layer (Type)	Output Shape	Parameters	Hyperparameters
Conv1d_1 (Conv1D)	(None, 38, 128)	5888	filter = 120, kernel_size = 3, strides = 1, padding = ‘valid’
Conv1d_2 (Conv1D)	(None, 36, 64)	24,640	filter = 64, kernel_size = 3, strides = 1, padding = ‘valid’
Conv1d_3 (Conv1D)	(None, 34, 32)	6176	filter = 32, kernel_size = 3, strides = 1, padding = ‘valid’
Dropout_1 (Dropout)	(None, 34, 32)	0	rate = 0.3
Max_pooling1d_1 (Max-pooling)	(None, 17, 32)	0	pool_size = 2, strides = None, padding = ‘valid’
Flatten_1 (Flatten)	(None, 544)	0	-
Dense_1 (Dense)	(None, 100)	54,500	units = 100, activation= RELU
Dense_2 (Dense)	(None, 5)	505	units = 5, activation = RELU
Total parameters: 91,709			
Trainable parameters: 91,709			
Non-trainable parameters: 0			

The loss function was set to ‘categorical_crossentropy’, while the optimizer was set to ‘adam’. Moreover, the batch size and the epoch were set to 32 and 600, respectively. Training the models took approximately 90 min per model by using Google Colaboratory, which is a cloud-based notebook environment that allows users to write, execute and share code in Google Drive. Google Colaboratory gives free access to GPUs (graphics processing unit) and TPUs (tensor processing unit) with the following characteristics and performance (Table 3).

**Table 3 animals-13-01886-t003:** Summary of the graphics processing unit (GPU) characteristics and performance made available in Google Colaboratory.

Parameter	Value
GPU	Nvidia K80/T4
GPU memory	12 GB/16 GB
GPU memory Clock	0.82 GHz/1.59 GHz
Performance	4.1 TFLOPS/8.1 TFLOPS
Support mixed precision	No/Yes
GPU release year	2014/2018
No. CPU cores	2
Available RAM	12 GB (upgradable to 26.75 GB)

CPU = central processing unit; RAM = random access memory.

Figure 4 reports the learning curve of the model, a line plot showing how the accuracy of the model increases over training time. Models are trained over a large number of epochs, allowing the learning algorithm to run until the error from the model has been sufficiently reduced. The epoch is a metric unit that indicates that each sample in the training dataset has had an opportunity to update the internal model parameters, and the number of epochs is a hyperparameter that provides the number of times that the learning algorithm will work through the entire training dataset [29].

### 2.5. Model Assessment

Average accuracy (macro and weighted), recall, precision and F1 score were calculated to measure the CNN’s capability in predicting cow behaviour [30]. Once the numbers of true positives (TP), true negatives (TN), false positives (FP) and false negatives (FN) have been set, the average accuracy is calculated as (TP + TN)/(TP + FP + FN + TN) and gives an overall measure of correctly identified behaviours [26]. Note that in average accuracy, all classes are assigned an equal weight when contributing their portion of the precision value to the total. This might not be a realistic calculation when there is a large amount of class imbalance. In the latter case, a weighted macro average is more informative. Weights are calculated by the frequency of the class in the truth column. The other parameters were calculated as follows: recall = TP/(TP + FN); precision = TP/(TP + FP); and F1 score = (2 × precision × recall)/(precision + recall). The latter is a single score that balances both the aspects of precision and recall in one number, as reported in the literature [30].

## 3. Results

Accuracy alone is considered to be an inappropriate measure of performance for imbalanced classification problems. Any model’s accuracy can often be overestimated due to the overwhelming number of cases. For this reason, the precision, recall and F1 score were calculated as well. The F1 score combines both precision and recall (sensitivity) in a single measurement that considers both properties and, in our case, confirms the value of the overall accuracy [30]. As reported in Table 4, the overall accuracy, macro and weighted average of precision, recall and F1 score of the model were all equal to, or higher than, 0.93 in the testing dataset. Among the specific behaviours (Table 4), resting showed the highest precision, recall and F1 score in both datasets, whereas standing still showed the lowest precision and F1 score. Moving was the behaviour with the lowest recall. The precision, recall and F1 score of single behaviours have ranges between 0.93 and 0.99. Table 5 shows the confusion matrix and thus the contingency table with two dimensions (‘Actual’ and ‘Predicted’) and identical sets of ‘classes’ in both dimensions.

The performance attained through the application of the CNN model in correctly classifying dairy cows’ behaviours greatly overperformed the classical ML models used (Figure 5). Figure 5 highlights an important difference (0.20 points) in the overall accuracy between CNN and RF, which was reported to be the best-working classical ML algorithm in behaviour detection [26]. The precision and sensitivity/recall scores are very similar in both the CNN and classical ML for resting behaviour. In behaviours that are quite similar with regard to their movement patterns, such as feeding, moving and standing still, the precision and sensitivity/recall scores are higher in the CNN by approximately 0.19–0.34 and 0.21–0.34 points, respectively. Regarding ruminating behaviour, the precision was very similar in both the CNN and RF with a difference of 0.12 points, whereas the difference for sensitivity/recall between the CNN and the highest value shown by the ML algorithms, in this case represented by XGB, was 0.33 points.

## 4. Discussion

In this study, we investigate the performance of a CNN model in classifying five behaviours of healthy dairy cows on the basis of tri-axial accelerometer data. DL models, including CNNs, have been known since the 1990s; however, in the beginning, they did not attract attention due to the lack of suitable computational resources and insufficient suitable and available data [20]. More recently, thanks to the widespread use of powerful computational resources and the availability of large amounts of suitable data, DL models have been applied to almost every domain, including animal health [2], individual recognition [21] and the application of computer vision systems to animal farming [23]. In the latter field, even pose estimation and locomotion pattern recognition achieved excellent results [24]. However, thus far, behaviour classification in dairy cows through the application of tri-axial accelerometers has been mainly performed with classical ML models, with few exceptions [16,17,18,19,22], whereas DL applications of accelerometry data were investigated in humans. Studies in humans showed an overall maximum F1 score of 0.97 when predicting daily living activities when using the LSTM model and multiple sensors [20] and of 0.82 when using an RNN and a single accelerometer [25].

The CNN model’s performance was compared with that obtained from the same raw data and using classical ML models [26] by using common metrics such as precision, sensitivity/recall and overall accuracy. Sensitivity and recall are used in ML and DL, respectively, but are calculated in the same way (sensitivity or recall = TP/(TP + FN)). Accuracy, although our data were imbalanced with regard to different behaviours, was used in the comparison between DL and classical ML models since it yielded the same results as the F1 score and was considered suitable.

Feeding, standing still and moving were among the most difficult behaviours to distinguish between, especially with the classical ML model, as also reported by Wang et al. [31]. These authors, by using a single accelerometer coupled with an adapting boosting algorithm, found low sensitivity and precision for both feeding (0.52 and 0.55) and standing still (0.46, 0.58) compared with other behaviours such as lying down (0.83, 0.82). During all these behaviours, the cows are standing and can move their head and body, and, in the cases of moving and feeding, they can also move their legs for small changes in position [15]. Moreover, during these behaviours, the sensor was similarly oriented along the 3 axes [20,26], which then changed when resting or ruminating in a lying position [15,26].

The highest accuracy of the CNN model, and of the other DL models [2,23,24], is favoured by their structure. These algorithms are made up of simple units organised in layers that are stacked to form ‘deep networks’ [24]. The links between units can be trained on the original data from the entire dataset and can learn to extract information by adding nonlinearity in the feature space, which is usually overlooked during manual feature extraction [20,24]. In particular, the CNN model has likely learned the nonlinear temporal dynamics of the predictors derived by the X, Y and Z acceleration values, an outcome that is not possible with conventional prediction models, as also reported by Con et al. [32] regarding deep learning models for clinical prediction in human diseases. On the other hand, even the best of the tested ML models, such as the RF for feeding, moving and standing still behaviours, and the XGB for resting and ruminating, showed lower accuracy compared with the CNN. This is likely due to some intrinsic characteristics of the algorithms. Both RF and XGB have often been used for the prediction of events such as the onset of a disease [4] or the performance of a behaviour [9] and show elements that enable high performance. RF can achieve high accuracy without hyperparameter optimization and has a lower probability of overfitting than decision trees. Cabezas et al. [33], using an accelerometer on a collar and RF, demonstrated sensitivity ranging from 0.58 for standing still to 0.94 for grazing. XGB is very efficient in capturing nonlinear relationships in the data due to many system and algorithmic optimizations, such as tree pruning, parallelization and cross-validation. In a previous study [9], for example, the XGB associated with an accelerometer on a collar was able to reach a sensitivity ranging from 0.82 for resting in a lying position, to 0.99 for grazing. Nevertheless, both RF and XGB, also likely due to the need for previous manual feature extraction, proved to be less adaptable than CNN in managing time series classification tasks involving large datasets.

The high overall accuracy (0.96) of the CNN was obtained despite the rather high number of behaviours tested, which is thought to severely affect prediction reliability [9]. Furthermore, the high values of precision, recall and F1 score (from 0.93 to 0.99) in correctly classifying all the five behaviours considered were achieved by using a single sensor, which implies reducing costs and has fewer drawbacks for both the cow and the farmer compared with the application of multiple sensors [15,26]. As reported in Table 6, other studies, by using tri-axial accelerometers, achieved high overall accuracy in behaviour prediction despite the differences in the adopted protocols.

When comparing results of behaviour prediction from different studies, even when the number and type of sensors match (e.g., accelerometers), it must be taken into account that the precision, recall and F1 score of different behaviours are strongly affected by the sensor position on the animal [9,15]. Fitting the sensor on the neck favours the prediction of feeding and ruminating because those behaviours are characterised by movements of the head, but it hampers the recognition of behaviours characterised by body position, such as resting. Moving the accelerometer from the neck to the leg when measuring standing, for example, raised sensitivity from 0.65 to 0.78 but when measuring feeding, decreased sensitivity from 0.96 to 0.81 [15]. The number of behaviours predicted affects the outcomes as well, since it is easier to find an algorithm fitted for one or few behaviours than one that performs well for multiple behaviours. Some studies, in fact, focused only on a maximum of three behaviours, either feeding, lying and standing [34] or feeding, ruminating and other behaviours [35].

Our findings, especially those obtained with DL, were more accurate or precise than those of previous studies reporting multiple behaviour predictions with the use of a single sensor and classical ML algorithms, with the exception of Riaboff et al. [9]. Roland et al. [36] and Martiskainen et al. [37], for example, found overall accuracies of 0.71 and 0.69, respectively and in general lower precision and sensitivity/recall for all the specific behaviours investigated (Table 6). Riaboff et al. [9], measuring multiple behaviours with an accelerometer on the collar and XGB, achieved results in line with our findings, with the exception of resting in a lying position, which showed a lower sensitivity (0.82). Other studies achieved high overall accuracies through additional accelerometers or other sensors [15,22,38]. From a practical point of view, achieving a higher accuracy and precision in the determination of the time spent resting, feeding, ruminating and standing allows farmers to better observe real changes, thereby reducing the number of both false positives and false negatives regarding the alerts given for alleged incoming diseases. The presence of lameness, for example, has already been associated with variations in feeding and lying time in cows [3,8,9], and the onset of bovine respiratory disease has been related to a higher pattern of variation over time in daily rumination in beef cattle [12]. For this reason, the improvement of accuracy and precision in the detection of time spent by healthy cows ruminating, feeding, resting and moving represents the necessary basis to improve the accuracy and precision of disease prediction. Moreover, the cows’ overall time budget can be associated with housing, management and feeding issues, such as resting area design, overstocking, ration formulation and delivery [39]. For example, assuming an average daily rumination time of 500 min, our CNN model, with an F1 score of 0.97, would yield an error of 15 min, whereas XGB, using the same raw data with a balanced accuracy of 0.805, would yield an error of 97.5 min.

The improvement in performance attained in this study, namely the switch from classical ML models to CNN, suggests that applying DL models might improve overall performance in observing the behaviours of both cows and animals in general by using accelerometry data. Similar outcomes have been reached in the detection of subclinical mastitis from milk composition data [2] and even in the prediction of inflammatory bowel diseases in humans [32]. On the other hand, Aways et al. [20] found only a slight improvement in the recognition of human activity with a DL model when compared with the classical ML models and suggested that when a performance plateau was reached (in this case, the F1 score was 0.97), further improvements might not be achievable, regardless of whether ML or DL models are used. Among the major drawbacks of the DL models is their ‘black box’ approach, which makes it impossible to produce a causal link between predictors and results [32]. Another drawback is the need for a large amount of labelled data, suitable software and powerful computational resources [32], even though the increasing application of new sensors and technology in the farming sector should fulfil this need.

## 5. Conclusions

To conclude, the application of a CNN model to data acquired through a single tri-axial accelerometer on mid-lactating dairy cows showed an overall high performance in successfully predicting multiple behaviours. The CNN model outperformed the outcomes previously obtained by the application of classical ML models to the same raw data. These results demonstrate the huge potential of DL in precision livestock farming applications. Although the results look promising, before applying these methods to commercial farms, it might be necessary to verify their reliability with regard to different breeds, farming conditions and sensors. Regarding the potential 24/7 application of the DL model, although transitional behaviours were not included in the behaviours of interest, this should not significantly affect overall accuracy since transition behaviours represent a negligible amount of time. Furthermore, the transition from one behaviour to another can be implicitly calculated by counting the number of transitions between the different behaviours.

## Figures and Tables

**Figure 1 animals-13-01886-f001:**
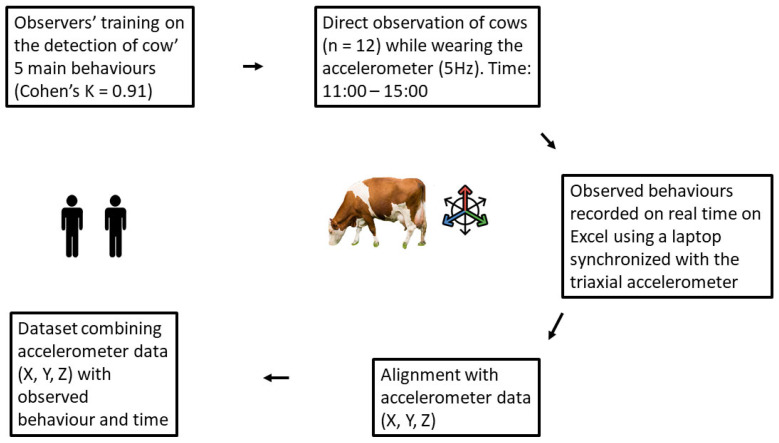
Data collection flow.

**Figure 2 animals-13-01886-f002:**
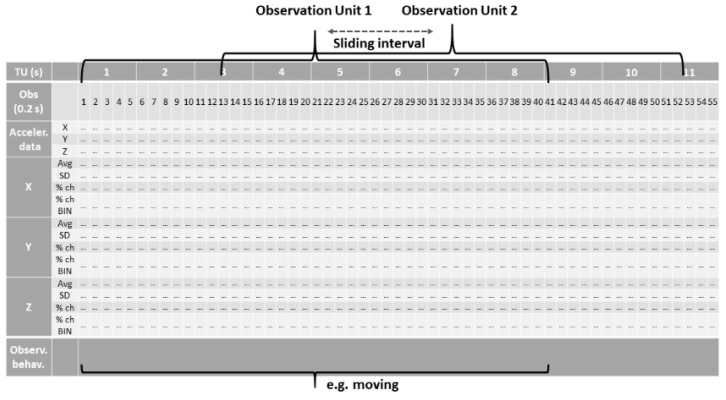
Graphical representation of the database metrics. TU = temporal unit (expressed in s); Obs = observation (every 0.2 s); Acceler. Data = raw acceleration data; X, Y, Z = acceleration value of each axis; AVG = average of the acceleration values within each observation unit; SD = standard deviation within each observation unit; % ch = percentage change between one observation and the previous one; % ch BIN = binary value related to the % change (if the percentage change is negative, the value given is 0; otherwise it is 1); Observ. Behav. = observed behaviour.

**Figure 3 animals-13-01886-f003:**
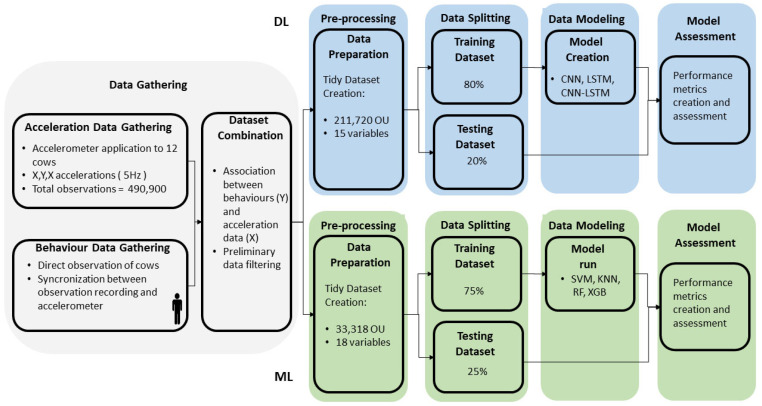
Schematic data flow and modelling. CNN = Convolutional neural network; DL = Deep learning; KNN = K Nearest Neighbours; ML = classical Machine learning; OU = Observation units; RF = Random Forest (RF), SVM = Support Vector Machine; XGB = Extreme Gradient Boosting.

**Figure 4 animals-13-01886-f004:**
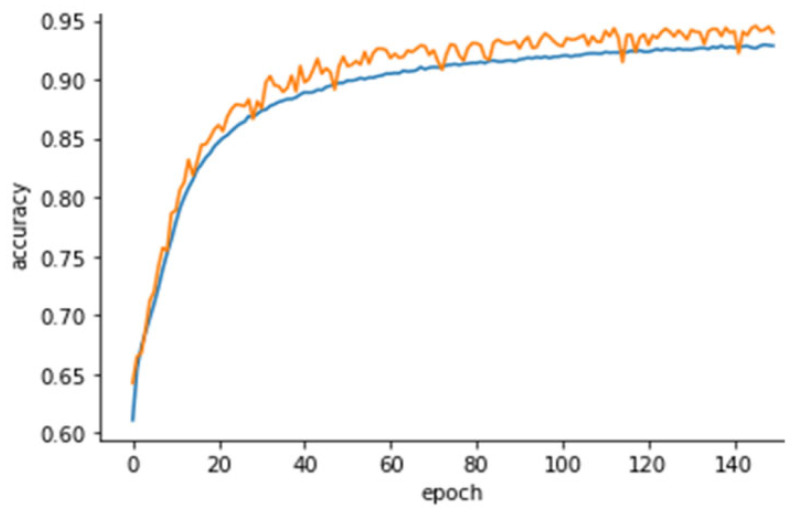
Learning curve of the CNN model showing the accuracy of the model considering the training (orange line) and test dataset (blue line) over the number of epochs.

**Figure 5 animals-13-01886-f005:**
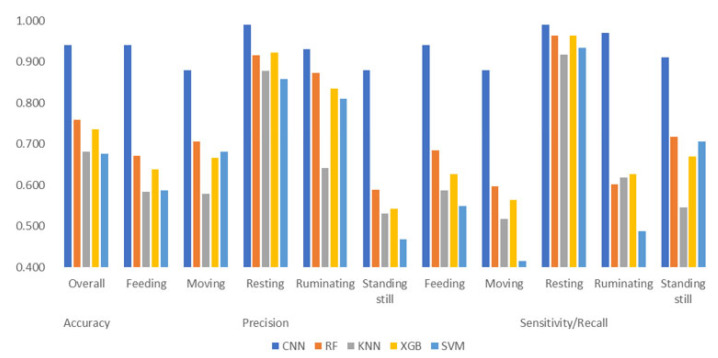
Overall accuracy, precision and sensitivity/recall of Convolutional Neural Network (CNN), Random Forest (RF), K Nearest Neighbours (KNN), Extreme Gradient Boosting (XGB) and Support Vector Machine (SVM) models in correctly classifying 5 behaviours. The performance of RF, KNN, XGB and SVM, which are classical ML models, is reported by Balasso et al. [26] for the same raw data used in this paper.

**Table 1 animals-13-01886-t001:** Behaviour descriptions of dairy cows.

Behaviour	Definition ^1^
Standing still	Cows stand still without moving their legs or showing any sign of activity
Feeding	Cows ingest feed and chew it at the feed bunk
Moving (walking or moving slightly)	Cows walk across the pen or, while standing, perform other behaviours other than those described here, such as agonistic behaviours and drinking, which are characterized by at least one step every 10 s
Ruminating	Cows chew a bolus, a process which begins upon regurgitating the bolus and ends when the bolus is swallowed, in either a standing or lying position
Resting	Cows lie on the floor, neither moving nor ruminating

^1^ Adapted from Balasso et al. [26].

**Table 4 animals-13-01886-t004:** Assessment of the accuracy, precision, recall and F1 score of the CNN and CNN-LSTM (values in brackets) models in the prediction of multiple behaviours in the testing dataset.

Behaviour	Precision	Recall	F1 Score	Number of Observation Units
Feeding	0.96 (0.89)	0.96 (0.91)	0.96 (0.90)	8192
Moving	0.94 (0.86)	0.94 (0.89)	0.94 (0.88)	8857
Resting	0.99 (0.98)	0.99 (0.96)	0.99 (0.97)	13,141
Ruminating	0.99 (0.88)	0.96 (0.92)	0.97 (0.90)	5001
Standing still	0.93 (0.88)	0.93 (0.83)	0.93 (0.85)	7144
Metrics				
Accuracy			0.96 (0.91)	42,335
Macro average	0.96 (0.90)	0.96 (0.90)	0.96 (0.90)	42,335
Weighted average	0.96 (0.91)	0.96 (0.91)	0.96 (0.91)	42,335

**Table 5 animals-13-01886-t005:** Confusion Matrix of the CNN model for the prediction of multiple behaviours.

Predicted	Actual				
	Feeding	Moving	Resting	Ruminating	Standing Still
Feeding	7896	147	12	1	136
Moving	185	8352	21	7	292
Resting	12	18	13,058	43	10
Ruminating	20	23	96	4793	69
Standing still	138	337	9	19	6641

**Table 6 animals-13-01886-t006:** Year, behaviour, number of animals (N), total sampling hours (h), number of accelerometers (Acceler.), use of other sensors, sensor location, acceleration sampling rate, models used and overall accuracy of the reported studies [Reference number] using tri-axial accelerometer to predict behaviour in cattle.

Study	Year	Behaviour	N	h	Acceler.	Other Sensors	Sensor Location	Sampling Rate (Hz)	Models	Overall Accuracy
Present	2023	F, M, R, Ru, Ss	12	27	Single	-	Left flank	5	**CNN**, KNN, LSTM, RF, SVM, XGB	0.96
[9]	2020	G, R, Ru, W	86	57	Single	-	Neck	59.5	ADA, RF, SVM, **XGB**,	0.98
[15]	2019	F, L, S	16	96	Multiple	-	Leg and neck	1	KNN, NB, **SVM**	0.98
[18]	2019	C, F, H, L, Li, M, Ru	6	68	Single	IMU	Neck	20	CNN, **LSTM-RNN**	0.89
[19]	2020	Cb	3	150	Single	IMU	Neck	20	LSTM-RNN	0.80
[22]	2022	F, L, Li, Ri, Ru	12	1066	Single	IMU	Neck	10	**Bid. LSTM**, LSTM-RNN	0.95
[26]	2022	F, L, M, R, Ru, S, Ss	12	27	Single	-	Left flank	5	KNN, **RF**, SVM, XGB	0.76
[31]	2018	C, F, L, S, W	5	200	Multiple	RFL	Legs and neck	1	MBP-ADAB	0.73
[33]	2022	G, L, O, Ru, Ss	10	50	Single	GPS	Neck	10	RF	0.88–0.93
[34]	2015	C, F, L, S, W	6	33	Single	-	Neck	50	**DT**, HMM, K-means, SVM	0.88, 0.82 **
[35]	2022	F, O, R	18	60	Single	PC	Neck and halter	10	HMM, **LDA**, PLSDA	0.83
[36]	2018	F, L, Ru, W	15	60	Single	-	Ear	10	HMM	0.71
[37]	2009	C, F, L Ru, S	30	95.5	Single	-	Neck	10	SVM	0.78
[38]	2023	D, F, R, Ru	30	156	Single	UWB	Neck	2	**DT**, HMM, K-means, SVM	0.99 *

Behaviours: Calving-related behaviours (Cb), Changing position (C), Drinking (D), Feeding (F), Grazing (G), Headbutting (H), Licking (Li), Lying (L), Moving (M), Others (O), Resting (R), Rub itching (Ri), Ruminating (Ru), Standing (S), Standing still (Ss), Walking (W); Models: Adaboost (ADA), Bidirectional Long Short-Term Memory networks (bid. LSTM), Convolutional Neural Network (CNN), Decision-Tree (DT), Extreme Gradient Boosting (XGB), Hidden Markov Model (HMM), K-means, K Nearest Neighbours (KNN), Linear Discriminant Analysis (LDA), Long Short-Term Memory networks (LSTM), Long Short-Term Memory–Recurrent Neural Network (LSTM-RNN), Multi-BP-AdaBoost (MBP-ADAB), Naive Bayes (NB), Partial Least-Squares Discriminant Analysis (PLSDA), Random Forest (RF), Support Vector Machine (SVM); Sensors: Global Positioning System (GPS), Inertial Measurement Unit (IMU), Radio Frequency Location (RFL), Pressure Changes (PC), Ultra-Wideband Location (UWB). Accuracy is referred to the best model (Bold). Where overall accuracy was missing it was replaced by * Coefficient of Determination (R2) or ** overall sensitivity and precision, respectively.

## Data Availability

The data presented in this study are available on request from the corresponding author. The data are not publicly available since they are still under analysis for further publications.

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
