# Peer review of "Uncovering Patterns in Dairy Cow Behaviour: A Deep Learning Approach with Tri-Axial Accelerometer Data"

_animals, 2023, doi:10.3390/ani13111886_

Round 1
Reviewer 1 Report
Comments:
The paper investigates the use of wearable sensors (accelerometers) for the classification of cattle behaviours, based on deep learning.
Some points need to be addressed in order to improve the paper quality before considering for publishing:
1. A major deficiency in the English expressions has been identified. Please have the paper polished by a professional English editor or native speaker. Please ensure that the numerous grammatical, expression, and punctuation errors are corrected throughout the manuscript.
2. The abstract should be rewritten: remove irrelevant information (e.g., details about the algorithm), objectives and novelties not clear, information is missing (e.g., considered behaviours)
3. Lines18-19, rephrase the sentence.
4. The sota in the introduction should be relevant for the objective of the study. Summarize the details about the use of sensors for mastitis/lameness/oestrus detection.
5. Lines 73-76: Deep learning were already used for behaviour classification using accelerometers, see:
L. Riaboff, L. Shalloo, A.F. Smeaton, S. Couvreur, A. Madouasse, M.T. Keane, Predicting livestock behaviour using accelerometers: A systematic review of processing techniques for ruminant behaviour prediction from raw accelerometer data, Computers and Electronics in Agriculture, Volume 192, 2022, 106610, https://doi.org/10.1016/j.compag.2021.106610.
6. Additional references should be included and discussed in the introduction:
Wu, Y.; Liu, M.; Peng, Z.; Liu, M.; Wang, M.; Peng, Y. Recognising Cattle Behaviour with Deep Residual Bidirectional LSTM Model Using a Wearable Movement Monitoring Collar. Agriculture 2022, 12, 1237. https://doi.org/10.3390/agriculture12081237
Peng, Y., Kondo, N., Fujiura, T., Suzuki, T., Wulandari, Yoshioka, H., Itoyama, E., 2019. Classification of multiple cattle behavior patterns using a recurrent neural network with long short-term memory and inertial measurement units. Comput. Electron. Agric. 157, 247–253.
Peng, Y., Kondo, N., Fujiura, T., Suzuki, T., Ouma, S., Wulandari, Yoshioka, H.,
Itoyama, E., 2020. Dam behavior patterns in Japanese black beef cattle prior to
calving: Automated detection using LSTM-RNN. Comput. Electron. Agric. 169,
105178. https://doi.org/10.1016/j.compag.2019.105178.
Da Silva, S.A., de Medeiros, V.W.C., Gonçalves, G.E., 2023. Monitoring and classification of cattle behavior: a survey. Smart Agricultural Technology 3, 10009
S. Benaissa, F.A.M. Tuyttens, D. Plets, L. Martens, L. Vandaele, W. Joseph, B. Sonck, Improved cattle behaviour monitoring by combining Ultra-Wideband location and accelerometer data, animal, Volume 17, Issue 4, 2023, 100730, https://doi.org/10.1016/j.animal.2023.100730.
7. The novelties of the study are not stated in the introduction. What is new in the study compared to previous studies?
8. Table 1. What about the other behaviours: drinking, agonistic behaviours? Were they included or excluded from the analysis?
9. Is the number of cows (12) representative? Why only 2 hours per day?
10. Lines 142-145: Using the individual accelerometer axes (X, Y, and Z) for the classification assumes that the sensor stays at the same position. In real scenarios, the collar (and sensor) is continuously moving around the cow’s neck.
11. Line 169: why excluding these intervals? Why not consider the behaviour with the longest duration/ highest percentage? Assume you have an interval where 80% is ruminating and 20% is resting, then this interval should be considered as ruminating.
12. Results: A confusion matrix could give more information about the detection performance.
13. The link between the performance in term of accuracy, precision, etc and the actual error in minutes is not presented.
Assume one ML algorithm has an accuracy of 95% and a DL algorithm has 97% for ruminating behaviour. What is the daily error in minutes for the two algorithms, taking in consideration a daily ruminating time of 8-10 hours?
14. Discussion section: add a comparison table containing the previous studies (compared to your study) including:
· Year of the study,
· The considered behaviours,
· Number of animals included,
· Dataset size (e.g., hours),
· Sensor location (neck, leg, tail, etc),
· Samling rate,
· Classification models,
· Performance (accuracy, precision, sensitivity, etc),
15. Line 416: Why specific “mid-lactating dairy cows”?
Author Response
Reply to Editor and reviewers
Title: Uncovering Patterns in Dairy Cow Behaviour: A Deep Learning Approach with Tri-axial Accelerometer Data
Authors’ reply: the authors thank the editor and reviewers who, through their remarks and requests, helped them to improve the quality of the manuscript.
Reviewer 1 (Corrections in the text are highlighted in Green)
Comments and Suggestions for Authors
Comments:
The paper investigates the use of wearable sensors (accelerometers) for the classification of cattle behaviours, based on deep learning.
Some points need to be addressed in order to improve the paper quality before considering for publishing:
- A major deficiency in the English expressions has been identified. Please have the paper polished by a professional English editor or native speaker. Please ensure that the numerous grammatical, expression, and punctuation errors are corrected throughout the manuscript.
Authors’ reply: The paper has been edited for proper English language, grammar, punctuation, spelling, and overall style by a professional editing service. All the changes are highlighted in yellow in the text.
- The abstract should be rewritten: remove irrelevant information (e.g., details about the algorithm), objectives and novelties not clear, information is missing (e.g., considered behaviours)
Authors’ reply: Done, L21-23, 24, 26-29.
- Lines18-19, rephrase the sentence.
Authors’ reply: Done, L 17-18
- The sota in the introduction should be relevant for the objective of the study. Summarize the details about the use of sensors for mastitis/lameness/oestrus detection.
Authors’ reply: We summarized the details as requested (L43, 50-51 and 55).
- Lines 73-76: Deep learning were already used for behaviour classification using accelerometers, see:
- Riaboff, L. Shalloo, A.F. Smeaton, S. Couvreur, A. Madouasse, M.T. Keane, Predicting livestock behaviour using accelerometers: A systematic review of processing techniques for ruminant behaviour prediction from raw accelerometer data, Computers and Electronics in Agriculture, Volume 192, 2022, 106610, https://doi.org/10.1016/j.compag.2021.106610.
- Additional references should be included and discussed in the introduction:
Wu, Y.; Liu, M.; Peng, Z.; Liu, M.; Wang, M.; Peng, Y. Recognising Cattle Behaviour with Deep Residual Bidirectional LSTM Model Using a Wearable Movement Monitoring Collar. Agriculture 2022, 12, 1237. https://doi.org/10.3390/agriculture12081237
Peng, Y., Kondo, N., Fujiura, T., Suzuki, T., Wulandari, Yoshioka, H., Itoyama, E., 2019. Classification of multiple cattle behavior patterns using a recurrent neural network with long short-term memory and inertial measurement units. Comput. Electron. Agric. 157, 247–253.
Peng, Y., Kondo, N., Fujiura, T., Suzuki, T., Ouma, S., Wulandari, Yoshioka, H.,
Itoyama, E., 2020. Dam behavior patterns in Japanese black beef cattle prior to
calving: Automated detection using LSTM-RNN. Comput. Electron. Agric. 169,
- https://doi.org/10.1016/j.compag.2019.105178.
Da Silva, S.A., de Medeiros, V.W.C., Gonçalves, G.E., 2023. Monitoring and classification of cattle behavior: a survey. Smart Agricultural Technology 3, 10009
- Benaissa, F.A.M. Tuyttens, D. Plets, L. Martens, L. Vandaele, W. Joseph, B. Sonck, Improved cattle behaviour monitoring by combining Ultra-Wideband location and accelerometer data, animal, Volume 17, Issue 4, 2023, 100730, https://doi.org/10.1016/j.animal.2023.100730.
Authors’ reply: We modified the sentence and added the references suggested (L 68, 86, 98-9). Moreover we referred to the above papers in the discussion (L309-310, 356-357, 392-393), in Table (6) and at lines 494-506, 512-514, 554-556.
- The novelties of the study are not stated in the introduction. What is new in the study compared to previous studies?
Authors’ reply: this study is the first to compare directly the performance in predicting dairy cows’ behaviour of ML models with Deep learning models using the same raw data acquired from an accelerometer (L98-102). As reported also by Riaboff et al., 2022 and Da Silva et al. 2023, overall there are only few studies using DL to predict cows’ behaviors starting from accelerometer data and none of them compared their performance to those of ML models.
- Table 1. What about the other behaviours: drinking, agonistic behaviours? Were they included or excluded from the analysis?
Authors’ reply: We focused on the main behaviours considered to be useful in the detection of the onset of the more impacting diseases as reported in different papers (Da Silva Santos et al., 2023; Riaboff et al., 2022; Peng et al., 2020). The other behaviours which included movement such as agonistic behaviours and drinking were included in the “moving” behaviour. We added it in Table 1.
- Is the number of cows (12) representative? Why only 2 hours per day?
Authors’ reply: As also reported in Table 6, twelve cows is a limited number, but it is in line with other studies made on this subject: [19] n=3; [31] n =5; [18, 34] n =6; [33] n = 10; [22, 26] n = 12); [15, 36] n = 16 and 15. Similar numbers were also used in human studies testing DL models to predict daily living activities. Furthermore, 12 is considered to be the minimum reliable sample size in a farm needed to find differences in behaviour (Havstad, 1991; range 5-20) and multiple parameters obtained through statistical power analysis, as reported by Tharangani et al. 2020 (doi:10.3390/ani10081363).
As regards the number of hours per day, since the aim was to check the performance in predicting the main 5 behaviours described in the paper, 2h in the late-morning and early afternoon, resulted enough to sample all the behaviours in each cow.
- Lines 142-145: Using the individual accelerometer axes (X, Y, and Z) for the classification assumes that the sensor stays at the same position. In real scenarios, the collar (and sensor) is continuously moving around the cow’s neck.
Authors’ reply: the sensor, as described at L 128-130 and in [26], was applied in the centre of the left flank paralumbar fossa through an elastic band ad kept in position by glue. For this reason, the sensor stood in the same position.
- Line 169: why excluding these intervals? Why not consider the behaviour with the longest duration/ highest percentage? Assume you have an interval where 80% is ruminating and 20% is resting, then this interval should be considered as ruminating.
Authors’ reply:
As regards the determination of the behaviour in a 8s interval when two behaviours are overlapping, we did exactly what you suggested. We only excluded observations for which the observed behaviour was not certain (L 148). The sentence at lines 158 was deleted since it was misleading.
- Results: A confusion matrix could give more information about the detection performance.
Authors’ reply: Done (Table 5). Since previously we hadn’t estimated the confusion matrix, to obtain it we had to run again the model in Google Colaboratory. This led to slightly different performance because the seed was saved. Thus, the training and validation test have been modified and so the performance, which were updated in the text and tables.
- The link between the performance in term of accuracy, precision, etc and the actual error in minutes is not presented.
Assume one ML algorithm has an accuracy of 95% and a DL algorithm has 97% for ruminating behaviour. What is the daily error in minutes for the two algorithms, taking in consideration a daily ruminating time of 8-10 hours?
Authors’ reply: We added a sentence explaining in practical terms what a difference in accuracy might mean in terms of minutes attributed to a wrong behaviour (L 403-406).
- Discussion section: add a comparison table containing the previous studies (compared to your study) including:
- Year of the study,
- The considered behaviours,
- Number of animals included,
- Dataset size (e.g., hours),
- Sensor location (neck, leg, tail, etc),
- Samling rate,
- Classification models,
- Performance (accuracy, precision, sensitivity, etc),
Authors’ reply: Done (Table 6)
- Line 416: Why specific “mid-lactating dairy cows”?
Authors’ reply: We did it just to be precise and describe the conditions under which the trial was performed.
Reviewer 2 Report
This text reports on a study that aimed to assess the effectiveness of deep learning (DL) in detecting changes in the behavior of dairy cows through accelerometer data. The study used a tri-axial accelerometer to detect five main behaviors in 12 cows, and the data was analyzed using classical machine learning (ML) algorithms and an 8-layer convolutional neural network (CNN).
The results showed that the CNN outperformed classical ML algorithms, achieving an overall accuracy and F1-score of 0.94. The CNN's high performance in predicting multiple behaviors using a single accelerometer suggests its potential for precision livestock farming. However, the authors suggest that further studies are necessary to assess the CNN's performance under different conditions.
This study aimed to evaluate how well a CNN method can classify healthy dairy cows' behavior using data from a tri-axial accelerometer and compare it to the performance of classical ML models that used the same raw data.
The ethical statement in the text is satisfactory as the study was conducted in accordance with D.Lgs. 26/2014 and EU Directive 2010/63/EU concerning experiments on animals, as is usual and sufficient.
The main contribution of the study is the improvement of the possibility of early detection and treatment of sick cows, which makes the treatment more effective, as changes in behavior precede signs of illness. The apathetic state of the animal is reflected in the disruption of the animal's behavior schedule.
Before considering the article for publication, certain aspects should be addressed:
1. I understand that the complexity of the experiment results in high time demands, but
including a larger number of animals would increase the representativeness of the results.
Why were only 12 cows chosen?
2. I consider the selected breed to be uncommon, so it would be appropriate to provide a brief
characterization of it (2-3 sentences).
3. Why was the monitoring conducted between 11 and 15 o'clock?
4. How were the types of behavior selected and why exactly 5 types of behavior and not more?
5. Each acronym should be explained upon first use.
6. Figure number 2, which includes a graphical representation of database metrics, has small
unreadable letters and is therefore not very comprehensible
The results of the experiment are dependable and the conclusion that the CNN model outperforms classical ML models can be considered confirmed.
The article brings new insights in its field, and I recommend its publication.
Author Response
Reply to Editor and reviewers
Title: Uncovering Patterns in Dairy Cow Behaviour: A Deep Learning Approach with Tri-axial Accelerometer Data
Authors’ reply: the authors thank the editor and reviewers who, through their remarks and requests, helped them to improve the quality of the manuscript.
Reviewer 2 (Corrections in the text are highlighted in turquoise)
This text reports on a study that aimed to assess the effectiveness of deep learning (DL) in detecting changes in the behavior of dairy cows through accelerometer data. The study used a tri-axial accelerometer to detect five main behaviors in 12 cows, and the data was analyzed using classical machine learning (ML) algorithms and an 8-layer convolutional neural network (CNN).
The results showed that the CNN outperformed classical ML algorithms, achieving an overall accuracy and F1-score of 0.94. The CNN's high performance in predicting multiple behaviors using a single accelerometer suggests its potential for precision livestock farming. However, the authors suggest that further studies are necessary to assess the CNN's performance under different conditions.
This study aimed to evaluate how well a CNN method can classify healthy dairy cows' behavior using data from a tri-axial accelerometer and compare it to the performance of classical ML models that used the same raw data.
The ethical statement in the text is satisfactory as the study was conducted in accordance with D.Lgs. 26/2014 and EU Directive 2010/63/EU concerning experiments on animals, as is usual and sufficient.
The main contribution of the study is the improvement of the possibility of early detection and treatment of sick cows, which makes the treatment more effective, as changes in behavior precede signs of illness. The apathetic state of the animal is reflected in the disruption of the animal's behavior schedule.
Before considering the article for publication, certain aspects should be addressed:
1. I understand that the complexity of the experiment results in high time demands, but
including a larger number of animals would increase the representativeness of the results.
Why were only 12 cows chosen?
Authors’ reply: As also reported in Table 6, twelve cows is a limited number, but it is in line with other studies made on this subject: [19] n=3; [31] n =5; [18, 34] n =6; [33] n = 10; [22, 26] n = 12); [15, 36] n = 16 and 15. Similar numbers were also used in human studies testing DL models to predict daily living activities. Furthermore, 12 is considered to be the minimum reliable sample size in a farm needed to find differences in behaviour (Havstad, 1991; range 5-20) and multiple parameters obtained through statistical power analysis, as reported by Tharangani et al. 2020 (doi:10.3390/ani10081363).
I consider the selected breed to be uncommon, so it would be appropriate to provide a brief
characterization of it (2-3 sentences).
Authors’ reply: Some information on the breed were added at lines (120-122).
Why was the monitoring conducted between 11 and 15 o'clock?
Authors’ reply: After performing preliminary observations done before the beginning of the research, we saw that during this time interval cows were performing all the behaviour of interest (L124-126)
How were the types of behavior selected and why exactly 5 types of behavior and not more?
Authors’ reply: We focused on the main behaviours considered to be useful in the detection of the onset of the more impacting diseases as reported in different papers (Da Silva Santos et al., 2023; Riaboff et al., 2022; Peng et al., 2020) and in Table 6.
Each acronym should be explained upon first use.
Authors’ reply: Done (L314).
Figure number 2, which includes a graphical representation of database metrics, has small
unreadable letters and is therefore not very comprehensible
Authors’ reply: The size of the letters has been increased (Figure 2)
The results of the experiment are dependable and the conclusion that the CNN model outperforms classical ML models can be considered confirmed.
Authors’ reply: many thanks to the reviewer
The article brings new insights in its field, and I recommend its publication.
Authors’ reply: many thanks to the reviewer
Round 2
Reviewer 1 Report
No more comments.